# Nurses’ Silence: Understanding the Impacts of Second Victim Phenomenon among Israeli Nurses

**DOI:** 10.3390/healthcare11131961

**Published:** 2023-07-07

**Authors:** Rinat Cohen, Yael Sela, Inbal Halevi Hochwald, Rachel Nissanholz-Gannot

**Affiliations:** 1Department of Health Systems Management, Ariel University, Ariel 4076405, Israel; rachelng@ariel.ac.il; 2Nursing Department, Ramat Gan Academic College, Ramat Gan 5211401, Israel; 3Nursing Department, Faculty of Social and Community Sciences, Ruppin Academic Center, Emeq-Hefer 4025000, Israel; yaels@ruppin.ac.il; 4School of Nursing, Max Stern Yezreel Valley College, The Yezreel Valley, Emeq Yezreel 1930600, Israel; inbalh@yvc.ac.il; 5Smokler Center for Health Policy Research, Meyers-JDC-Brookdale Institute, Jerusalem 9103702, Israel

**Keywords:** second victim phenomenon, nurse, organizational support, barriers, quality of care

## Abstract

Introduction: The ‘second victim’ phenomenon, (SVP) refers to a health professional who was involved in an adverse event (AE) and continues to suffer from the event to the detriment of personal and professional functioning. The second victims’ natural history of recovery model predicts stages of the phenomenon from AE occurrence until the ‘moving on’ stage and serves as a suitable structure for many organizational support programs worldwide. Purpose: Using the second victims’ natural history of recovery model to examine the impact of the SVP on Israeli nurses, with a specific focus on the organizational support they felt they required compared with the support they felt that they had received from their organizations. Methods: Fifteen in-depth interviews were conducted, using a semi-structured questionnaire, among nurses who had experienced the SVP. The interviews were recorded subject to the interviewees’ consent, transcribed, and analyzed using thematic content analysis. Findings: Throughout all six stages of recovery, all interviewees reported physical and emotional manifestations following exposure to an AE, regardless of the type of event or severity. They also reported difficulty in emotion regulation, as well as damage to functioning and overall quality of life. Most of the nurse interviewees reported a need to share the events with someone, but, despite this desire to receive appropriate support, almost none of them proactively requested help from a professional source, nor did their organizational management initiate proactive support. This lack of referral for further assistance is possibly explained through limited awareness of the SVP as a valid response to an AE, a perceived lack of legitimacy to receive organizational support, and personal barriers that accompany the phenomenon. Conclusions: Appropriate organizational support, offered proximal to an AE as well as over time, is essential for the nurse, the patient, and the organization. Personal barriers, together with limited awareness, may challenge the identification and provision of appropriate assistance. Hence, it is important to address the phenomenon as part of the general organizational policy to improve the quality of care and patient safety.

## 1. Introduction

Medical treatments that result in unintended harm or negative outcomes for the patient are commonly referred to as adverse events (AEs) and affect at least one in ten patients worldwide [1,2]. While many in the healthcare system have acknowledged that making mistakes is an inherent aspect of human nature [3], and even the most skilled professionals can make mistakes, still, healthcare providers struggle with the negative impact and repercussions of AE exposure. Over the past two decades, it has been increasingly recognized that providers who witness or participate in an AE, even without making any error or causing harm, may suffer adverse effects [4,5,6,7], an occurrence often referred to as the ‘second victim phenomenon’ (SVP) [4,5,6]. Recently, an international group of experts agreed on a consensus definition of the second victim as ‘any healthcare worker who is directly or indirectly involved in an unanticipated adverse patient event, unintentional healthcare error, or patient injury, and who becomes victimized in the sense that they are also negatively impacted’ ([8] p. 6).

Healthcare providers suffering from the SVP have negative emotional responses such as guilt, shame, anxiety, and anger, which may lead to a reduced sense of professional confidence and increasing isolation from peers and supervisors. These reactions may occur immediately or years after the event exposure [7,9,10], causing physical and mental health issues, including headaches, sleep disorders, eating disorders, depression, anxiety, and post-traumatic stress disorder [11,12,13,14,15]. Health professionals may turn to alcohol or drug use as a coping mechanism and may even attempt suicide. The impact of reduced professional confidence may lead to defensive practices, avoidance, and abandonment of the workplace and profession [5,11,15].

To create a comprehensive intervention to combat the SVP and assist providers suffering from these secondary effects, Scott et al. [7] developed the second victims’ natural history of recovery model, which is derived from six stages of a coping trajectory: stages include (1) chaos and accident response: quick incident review, providing immediate patient care and monitoring; (2) intrusive reflections: overwhelming, disturbing, and obsessive thoughts about the event; (3) restoring personal and professional integrity: assessing for organizational and family support; (4) enduring the inquisition: raising concerns regarding the institution’s responses such as a job or license loss and/or future lawsuit; (5) obtaining emotional first aid: seeking first emotional support from family, peers, and supervisors; and (6) moving on stage: thriving against surviving, accompanied by disturbing thoughts and sadness, or dropping out.

The second victims’ natural history of recovery model is a valuable framework for healthcare organizations to utilize to establish support programs and offer appropriate staff resources. However, despite the high prevalence and possible severity of the effect of the SVP on both the provider and subsequent quality of care, many healthcare systems worldwide lack awareness of the phenomenon and fail to provide the necessary resources to manage it [16,17,18]. In Israel, there have been limited studies conducted on the SVP in general, and with nurses specifically. One mixed methods study analyzed responses of nurses working in internal medicine departments, to understand the effects of patients’ suicidal attempts and explore the association between these experiences and nurse absenteeism and turnover. Seven nurses contributed to the qualitative analysis and identified feelings of isolation, helplessness, and loneliness after the incident. Within the quantitative analysis, the researchers identified a correlation between second victim distress, loneliness, absenteeism, and turnover. Higher levels of emotional distress were also negatively correlated with organizational support [18].

The other study to investigate the SVP in Israel compared nurses’ responses to making a medication error at two points in time (2005 compared with 2018). Researchers found that when the organizational risk management team took a non-blameful approach to errors, more positive second victim functioning was found. However, responses to the mental anguish, fear, and emotional impact over time were similar among nurses from all types of organizations, not just within those organizations that took a non-blameful approach [19]. These studies demonstrate the real physical and emotional toll that AEs can have on nurses in Israel, as well as demonstrate a need for healthcare organizations to recognize the impact of the SVP and to provide appropriate support to affected providers.

The objective of this study was to examine the impact of the SVP on Israeli nurses, focusing on the organizational support they felt they required, compared with the support they felt that they had received from their organizations, using the second victims’ natural history of recovery model as a guide [7].

## 2. Material and Methods

### 2.1. Study Design

This study utilized a descriptive qualitative approach to examine nurses’ experiences of the SVP after exposure to an AE, as defined by the second victims’ natural history of recovery model [7]. We sought to understand the organizational support provided to nurses in each of the six coping stages outlined in the model. Upon receipt of ethical approval of the participating academic institution (#AU-20220409), we announced the study on social networks and invited nurses who had experienced the SVP to participate in interviews. Recruitment was carried out by posting a request for interviews on a nurses’ social network, and 30 nurses expressed interest. Informed consent was obtained from all interviewees, and their anonymity and confidentiality were guaranteed. They were informed of their right to withdraw from the study at any time without consequences. All collected data were kept confidential on a password-protected computer. Participants could choose to be recorded or not.

### 2.2. Participants and Recruitment:

We used purposive sampling to select nurses who had experienced the SVP for this descriptive qualitative study. We targeted participants with relevant knowledge and experience regarding the study’s aim of investigating nurses’ perceptions, behaviors, attitudes, and coping strategies regarding the SVP. The sample size was based on data saturation until no new information was obtained from further participants [20].

### 2.3. Research Process

Nurses were interviewed between December 2022 and February 2023. The interviews were between 60 and 90 min long. Prior to the interviews, participants were presented with a broad definition of the SVP and given time to recall their experiences. Data collection followed an iterative process of collecting, coding, and analyzing data [21]. All interviews were conducted by one researcher only. Eight interviews were recorded and transcribed, while seven additional interviews were conducted but not recorded (due to participants’ refusal) and analyzed using thematic analysis. The interviewer followed a previously developed guide [7]. Interviewees were asked to recall a significant incident they had experienced, regardless of how much time had passed since the event; they were also asked about the circumstances of a negative event, as well as the physical and psychosocial symptoms experienced during the event and throughout the recovery period. In addition, participants were asked about organizational support and recommendations for improving post-event support. This format allowed the researcher to understand the experiences of the provider alongside the desired organizational support when exposed to a negative event more deeply [7]. To increase reliability, the interviewer summarized each interview with the participant to check for and clarify any misconceptions or additional information.

### 2.4. Data Analysis

Data were analyzed using thematic analysis based on subcategories identified in previous studies [5,7]. In the first stage, the AEs mentioned in the interviews were quantified and categorized according to incident type and whether they caused harm to the patient or not. Descriptive qualitative approaches such as content analysis and thematic analysis were employed to analyze the data [22]. In the second stage of the analysis, transcripts were transcribed, coded, and analyzed manually to identify meaningful statements and compress the meaning units. Content analysis was conducted on the transcripts to comprehend the experiences, emotions, and expressions of the phenomenon, as well as the level and the quality of the organizational support received relative to the desired one. All the participating researchers contributed to the final content analysis. Results were cross-checked to validate the accuracy, and member checks were carried out to ensure credibility [23].

## 3. Results

The study included 15 nurses who met the inclusion criteria (consisting of five men and 10 women), ages 30 to 57 years. Ten nurses held a Master’s degree in nursing, two nurses were doctoral candidates in health systems management, and three held Bachelor’s degrees; there was adequate representation from different health departments and organizations. The length of working experience varied from 1 to 25 years (Table 1).

All 15 participants reported experiencing at least one significant AE during their career that had made an impact on both their health and professional functioning (Table 2). Among them, five interviewees discussed an incident where they caused harm to a patient due to an error. Three interviewees reported an error without harm to the patient. Seven interviewees reported significant harm to patients without an error.

Our findings are represented with the following themes: (1) the second victims’ natural history of recovery [7]; and (2) the actual receipt of organizational support compared with the desired support.

### 3.1. Second Victims’ Natural History of Recovery

#### 3.1.1. Stage 1: ‘Chaos and Incidence Response’

All the interviewees expressed similar experiences after the AE exposure. Eleven participants revealed that they handled the situation by responding to the patient’s immediate needs and continuously monitoring his/her condition, as required. However, four participants disclosed that their professional performance was negatively affected, and they had to take a moment or temporarily relinquish their duties to other staff members.


*I couldn’t continue to function. I needed a break, go out, smoke, and talk to a friend on the phone for a moment.*
(ED nurse, age 35)


*I was unable to function, I asked someone to replace me.*
(ICU nurse, age 30)

Some nurses reported a turbulent emotional experience, which led to initial feelings of shock and stress, accompanied by disbelief, guilt, and shame. Some expressed experiencing intense physical sensations such as nausea, sweating, racing heart, hot flashes, headaches, and dizziness.


*As soon as I realized that I was wrong, I suddenly felt a wave of heat inside me. I became dizzy, I felt my heart beating in my throat. I felt nauseated, and my hands were shaking.*
(Pediatric surgical nurse, age 53)

#### 3.1.2. Stage 2: ‘Intrusive Reflections’

In the aftermath of the event, the nurses reported a continuation of emotional distress for days and even months. Some reported symptoms such as anxiety, depression, insomnia, and nightmares. Others also had intrusive thoughts related to the event, and concerns about the patient’s future, which made it difficult to manage their personal lives, including interaction with their children after work.


*I would put my head on the pillow and my thoughts would race. What had I done, how did it happen, what will happen now? I couldn’t fall asleep.*
(ED nurse, age 42)


*I get home and the kids are running around, eagerly wanting to play, and I keep thinking about what had just happened, I can’t get away.*
(ED nurse, age 42)


*Even though it’s been months since the event, I remember it in detail as if it were yesterday, I still can’t believe it happened to me. I want to cry when I think about it.*
(ED nurse, age 35)

Some interviewees attributed blame to the healthcare system and their work environment, feeling powerless and helpless following the event.


*How can they leave two nurses on 30 patients? Obviously, there will be falls we won’t be able to get to everyone.*
(General surgical nurse, age 32)

#### 3.1.3. Stage 3: ‘Restoring Personal Integrity’

The study found that participants expected support from trusted individuals, such as colleagues, supervisors, friends, or family, to restore their integrity. Three nurses unofficially contacted colleagues in their department for a ventilation call. Those who sought support reported that the conversation helped them cope with their emotions. Two had formal conversations with their direct manager and risk management team. They noted that the risk management team and management responded swiftly, providing support within a few hours after the AE, which greatly assisted in rebuilding their self-assurance, confidence, and sense of capability. Nevertheless, most (10 participants) expressed feelings of shame or embarrassment about admitting to negative emotions and needing assistance due to fear of rejection. They were concerned that their colleagues or supervisors might view them as incompetent, unprofessional, or weak. Some even reported that peer gossip hindered their ability to move on, leading to increased memories of the event, personal doubt, and professional insecurity.


*I don’t believe that support from someone outside can help me. I need someone who could empathize with me, acknowledge my situation, offer some encouraging words, and reassure me that I am still a competent nurse. I prefer to share with people who can understand me like colleagues in my department who know me and understand the situation.*
(PICU nurse, age 44)


*My direct manager was with me the whole way, I felt I had someone to trust.*
(Pediatric surgical nurse, age 53)


*I don’t know if I have anyone to turn to for help in my department, following the incident, my husband accompanied me to, a private psychiatrist and we commenced medication therapy.*
(ED nurse, age 35)

Six interviewees pointed out that they sometimes found it extremely difficult to seek support from their family and friends as these individuals were unable to fully grasp the complexity of their situation.


*I couldn’t tell my wife, nothing I’m the stronger one between us. It would just make her anxious.*
(ED nurse, age 42)

#### 3.1.4. Stage4: ‘Enduring the Inquisition’

During the investigation stage, direct management and a representative from the local risk management department were present. In this investigation, the nurses were asked about the incident, what caused it, the patient involved, and the actions taken during and after the event. They also discussed how to prevent similar situations from happening in the future. Some even were informed that the AE was because of organizational failure and that organizational lessons were learned. In some cases, a representative from the Ministry of Health was also present, increasing the pressure level.

The participants shared that reporting the incident and debriefing was highly challenging, and some even considered it traumatic. They approached the clarification meeting with difficult emotions that intensified during the meeting, resulting in persistent feelings of anxiety and restlessness. Most of the nurses were worried that their involvement in an AE could lead to disciplinary action or damage their reputation, including job security, licensure, and future litigation.


*All nurses know what AEs are, and how to fill out event reports…but they don’t always know what the consequences of these types of events are.*
(Pediatric oncology nurse, age 35)


*I live in constant anxiety, we don’t have the support of the head nurse, and we get a lot of anger when we make mistakes or when we do not meet her expectations.*
(Gynecology department nurse, age 50)


*I am in constant anxiety; I know I am alone in this war. You expect some information and answers, but that did not happen.*
(ED nurse, age 42)

During the interview, some participants were afraid of the consequences and emphasized the importance of maintaining anonymity to avoid any potential negative outcomes at work:


*I’m not comfortable talking about it... It makes me anxious; I don’t want exposure.*
(ED nurse, 35)

#### 3.1.5. Stage 5: ‘Obtaining Emotional First Aid’

Almost all participants [14] expressed a need for help. However, most of them did not apply for proactive help-seeking, mainly because of a lack of SVP awareness or lack of organizational support legitimacy. Thirteen nurses did not seek emotional support from the organization, while three engaged in seeking support. Two participants reached out to their direct manager and the risk management department, and one nurse consulted a private psychiatrist outside of the organization. Most of the participants expressed disappointment over the organization’s lack of proactive communication to clarify their emotional state during the incident and the investigation stage, which gave them a sense of being unheard and lonely. Some even assumed that a lack of SVP awareness shows that the organization prioritizes the needs of the patient and the organization over its employees’ needs. Three participants noted that their organization offers emotional consulting services since COVID-19, but they did not find it suitable for their needs.


*I have the feeling that we are not seen. There is no support that the organization provides.*
(Gynecology department nurse, age 50)


*I have no one to talk to, I don’t feel that there is anyone who cares about me or listen to me. No one asks how I feel, or how am I? Or if I have any questions.*
(Gynecology department nurse, 50)


*I am nobody in the big system, and I have no place to come and share.*
(ED nurse, age 35)

#### 3.1.6. Stage 6: ‘Moving on-Dropping out, Surviving, or Thriving’

After an AE exposure, nurses reported attempts to cope and move on with their lives with three possible consequences: dropping out, surviving, or thriving.

Six participants described thoughts of dropping out: the emotional aftermath of an AE led them to consider leaving the workplace or nursing profession altogether. Three interviewees eventually, as a result, changed their workplace following the incident.


*The team is constantly dropping out. … it is mainly, in my opinion, the lack of support. They (the management) don’t see us at all. Negative feelings engorge, and nobody cares.*
(Pediatric oncology nurse, age 35)

Four participants described the behavior of surviving. Nurses reported function reverting while facing emotional distress, struggling with negative thoughts, and struggling to find joy or fulfillment in their profession. They mentioned becoming more cautious in adhering to procedures, resulting in defensive medicine and/or over-treatment.


*I work defensively, sometimes also at the expense of extending waiting times and writing very long nursing reports, which creates conflicts with my peers.*
(ED nurse, age 42)


*I shortened a process and that’s why I was wrong, following the incident I never shortened processes again.*
(PICU nurse, age 44)

Loss of confidence in their ability to provide high-quality care and a decrease in professional self-efficacy led to avoidant behavior and empathy erosion.


*I feel insecure, I avoid treating complex patients.*
(General surgical nurse, age 32)


*Following an error, nurses experience shame towards both the child and their family, leading them to avoid making eye contact.*
(Pediatric oncology nurse, age 35)


*I became less empathetic. I grew the skin of an elephant.*
(ICU, age 50)

Two participants described the behavior of thriving. Nurses reported that they used the experience to grow and develop as professionals. With the assistance of organizational support, they felt it easy to discuss their experiences with their colleagues in a formal setting, aiming to provide others with a learning opportunity through their mistakes.


*The risk manager told me that everybody can make a mistake. It was so important for me to hear this sentence. I went with her and spoke about the case in many departments so that everyone will learn from my experience, and it wouldn’t happen again.*
(Pediatric surgical nurse, age 53)

### 3.2. Desired Organizational Support Compared with Received

The participants expressed that they lacked both general support and emotional support, with only two nurses reporting receiving appropriate organizational support. Most (11) of the nurses felt lonely and experienced feelings of concern and distress, with no one to turn to for support. As they mentioned, colleagues and superiors made little proactive effort to understand the emotional state of the nurses or address their needs. Across all six stages, the participants expected empathy, transparency, reliable information at the right time, and organizational support from colleagues or supervisors, but these expectations were not met.

They suggested ways to assist them in similar situations in the future. Table 3 displays the degree of organizational support that the participants reported as received compared with the desired level of organizational support at the corresponding stage.

The interviewees stressed the importance of spreading awareness about the SVP and the necessity of seeking support when needed.


*It is important to teach nursing students that to make a mistake is human and to ask for help without shame or feeling of weakness.*
(General surgical nurse, age 40)

They expressed a desire to receive training on the subject.


*The concept of a ‘second victim’ is not discussed or recognized in our organization. If there is any kind of emotional support, it is only in very rare moments of crisis. Very much a Band-Aid. No one uses it.*
(Pediatric oncology nurse, age 35)

Interviewees discussed their feelings of vulnerability and expressed the importance of receiving reassurance and support from colleagues and superiors, during the initial stages of the incident. They requested assurance that the patients were unharmed, expressed a need to feel seen and supported, and emphasized the significance of providing a clear and honest explanation of what to expect during the investigation and its potential consequences.


*I don’t make mistakes on purpose, the most important thing for me was to know that I didn’t harm the patient.*
(Pediatric oncology nurse, age 35)

During the later stages of the incident, they shared the importance of acknowledging the legitimacy of seeking professional support both within and outside of the organization, even as a requirement. Some participants recommended using anonymous sources of support to express their thoughts and emotions freely without the fear of negative consequences.


*I don’t feel comfortable sharing my feelings with the person in charge, or the staff. I prefer a complete separation.*
(ED nurse, age 35)

## 4. Discussion

In this study, we examined the experiences of 15 Israeli nurses that coped with the SVP due to an AE exposure. Regardless of the presence of medical error or patient harm, all the interviewees expressed similar experiences after the exposure in regards to coping with both immediate and long-term effects of the AE, as has been noted in previous studies among health professionals after an AE involvement [5,7,24,25]. In our study, the emotional and physical effects as well as the personal and professional consequences were described using Scott’s second victims’ natural history of recovery model [7]. Using this description, we tried to map out gaps between the organizational support that nurses received compared with the organizational support that was desired.

On discovering the AE, all the interviewees reported a physical stress response, such as shaking hands, heart palpitations, heat flashes, and nausea, as seen in other studies around the globe [4,5,19]. We also found a wide range of emotional reactions to the event, including a sense of guilt, shame, fear, anxiety, anger, and isolation. These feelings, accompanied by obsessive thinking about the event, have been identified among nurses in a range of specialties and incident types [18,19,24]. The development of the SVP was rooted in these negative emotions, while simultaneously hindering the reception of adequate organizational assistance [5,10]. This first discovery reaction of an AE is typical of the extreme stress response, as seen in many incidences of the first trauma reaction [4,26]. In our study, nurses expressed a desire and need to receive initial support and talk to someone right after the AE occurred. These types of support programs, which offer first aid within 12 h of the event through peer support or anonymous hotlines, have been implemented in several countries [24,27].

In the third and fourth stages, nurses were seeking a way to regain their professional competence, reporting fear of rejection, and expressing concerns over disciplinary action or job security. Similar findings have been observed in other studies [5,9,16,18,28,29]. Additionally, some studies have mentioned that a sense of isolation may have contributed to delays in receiving the necessary assistance to cope with the situation [16,18,30]. In our study, it appears that nurses genuinely wanted to discuss the AE with someone. Some interviewees even tried to talk to colleagues or a direct manager. However, the majority felt that they did not have anyone with whom to share. These responses are similar to those seen by Chan and colleagues, who found that staff needed a process of sharing their experiences to reduce the emotional burden and enhance the management of distress, especially when the conversation enabled future learning opportunities [10]. Their study also raised the need for peer support and mentoring as potential tools for a way to open up the topic of the SVP for discussion and analysis [10]. Some of our participants did seek out informal discussions or conversations with colleagues or direct supervisors but this was not through formal organizational sanctioned channels. The nurses’ expectations for adequate organizational support did not materialize in most cases, nor were there proactive requests from direct management or colleagues to provide this support. These feelings of isolation and the lack of organizational support as a predictor of absenteeism and dropping out were summarized in one of the few studies on the SVP conducted in Israel [18].

The fifth stage expressed obtaining emotional first aid. Most of the participants expressed the need for support and a desire to share their experiences and feelings with a person who could understand their situation and respond with empathy and confidentiality. Nevertheless, most of the nurses did nothing to request this desired support. Many participants suggested that they lacked SVP awareness or did not feel the legitimacy to request organizational support, accompanied by personal barriers such as a sense of guilt, shame, anxiety, and anger. Similar findings have been noted in other studies [5,7,10,18]. These barriers were particularly associated with the fear of the potential reaction of the organization [5,10]. Moreover, in our study, we found disagreements about the type of adequate support the nurses desire. Some requested peer support, while some preferred a direct manager’s support, and others would have chosen anonymous external professional support. Similar debates were reported in previous studies [5,10,18,19,24,30]. Several studies found that desired support should be provided by professionals [5]. Others, in contrast, suggested peer support [31,32]. Thus, support must be well-tailored according to the provider’s needs to increase compliance with treatment [33].

In the last phase, the nurses attempted various ways to move on with their life, expressing the consequences of chosen coping strategies, such as dropping out, surviving through their emotional distress, and returning to work but struggling to find joy or job fulfillment. Many expressed a loss of confidence in their ability to provide high-quality care accompanied by a sense of loneliness contributing to the thought of dropping out. When the organizational culture encourages punishment, or the nurse feels uncertain ‘under investigation’, the consequences of the phenomenon are intensified [30].

Throughout all six stages, most of the nurses described a need for support, but they did not consider the organization as a relevant resource. Nurses that had sought support and help from the health organization, expressed disappointment and distrust at all stages of the SVP recovery path. Maybe the paradox is the existence of a system that is expected to support the employee (second victim) while, at the same time, conducting an inquiry into the employee’s actions, and representing the best interests of itself (third victim), as well as those of the patient (first victim) [34].

### Limitations

This was an initial qualitative study of a topic not yet researched in Israel; we hope to pursue larger mixed methods studies in the future, as awareness of this phenomenon grows among healthcare professionals. One significant limitation was the nurses’ fear of revealing the occurrence of an AE and the personal response to an unknown researcher through a social network. To mitigate this limitation, we emphasized anonymity and confidentiality, and the interviewees were allowed to stop the interview at any time. Additionally, we gave participants the freedom to choose whether to record the interview; indeed, seven interviewees asked us not to record the interview. Data were collected from representatives of different health organizations to represent different populations. However, our findings may not necessarily apply to other nurses, other health organizations, or other countries.

## 5. Conclusions and Recommendations

The SVP is an important issue in healthcare that can negatively affect both patient safety and the well-being of healthcare providers involved in an AE. A holistic approach recognizes the interdependence of patients and healthcare professionals, and provides support and resources to providers affected by the phenomenon. Programs that have taken these aspects into consideration have proven successful in other countries. For example, in an anesthesiology department in the United States, a three-tiered support program was implemented; over 90 staff members requested some level of anonymous support. Of those who received support, the large majority felt that the help offered by trained peers had been very beneficial [35]. A descriptive study in Austria asked participants in one large healthcare facility to rank theoretical support options. The option chosen most often was receiving legal advice after an AE, followed by increased access to professional counseling [31]. In Israel, the nursing administration took the initiative to educate supervising staff about a more supportive approach to managing medication errors. Their analysis found that, overall, nursing staff felt more supported after such an event, and there was less fear of punishment. These results demonstrate the possibility of raising awareness and changing the organizational culture [19]. Developing organizational resources and implementing support programs for those experiencing the SVP within an organization, raising awareness of the SVP among management, and building dedicated training programs for providers and peers are all necessary actions.

## Figures and Tables

**Table 1 healthcare-11-01961-t001:** Sociodemographic characteristics of the interviewees.

Variables	Type	Amount
Gender	Male	5
	Female	10
Age		Mean 35 (30–57)
Education level	RN Ph.D. Students	2
	RN MA	10
	RN BA	3
Another professional position	Clinical instructors	6
	Academic lecturers	8
Professional seniority		Mean 15 (1–25)
Department	Pediatrician oncology	1
	General oncology	1
	Emergency Department (ED)	3
	Intensive Care Unit (ICU)	3
	Pediatric Intensive Care Unit (PICU)	2
	General surgery	2
	Pediatric surgery	1
	Gynecology	2

**Table 2 healthcare-11-01961-t002:** Summary of significant adverse events.

Types of Events	Details	Estimated Time since the Event at the Time of Interview
Five reports of significant damage to the patient following an error	A mistake in identifying a patient and giving drug treatment to the coordinating sibling. Administering drug therapy in the wrong way. Administering medication in the wrong dosage. Error in the blood test process. Administering drug therapy in a double dose.	Four years Two months Two months Six months Four years
Three reports of an error without harm to the patient	Error in the process of preparing the medication. Error in patient identification. Error in the process of fluids preparation.	One year One year Eight months
Seven reports of significant damage caused to the patient without the commission of an error	Recurrent patient falls × 2. Unexpected resuscitation of a child during which the nurse discovered potassium levels above 15 Meq. Unexpected death × 2. Exposure to violence × 2.	Two years Four years Ten months Two months

**Table 3 healthcare-11-01961-t003:** Organizational support that nurses reported as received compared with the desired.

Stage	Characteristics	Received Support	Desired Support
Chaos and incidence response	Immediate stabilization of the patient or shifting the patient’s treatment to a colleague. Overflow and severe physical and emotional shock reaction. Overflow of fear in coping with the patient’s family’s response. Documentation of an AE report upon event discovery.	In most cases [13] no proactive contact was made.	Emotional first aid, provided by colleagues or a direct manager.
Intrusive reflections	Recurring disturbing thoughts accompanied by physical and emotional disorders. Daily functioning disorders, decrease in professional functioning.	Mostly, organization, direct manager, or colleagues did not proactively contact nurses to clarify the need for emotional assistance or to maintain an explanation of what to expect.	Clear explanation of what to expect during the investigation stage and its consequences. Assurance that the patients were unharmed. Need to feel seen and supported.
Restoring personal integrity	A strong desire to receive support to restore self-integrity, accompanied by negative emotions and fear of professional rejection that increased the feeling of loneliness.	Mostly, no proactive contact was made to find out their emotional state.	Emotional support from colleagues and a direct manager.
Enduring the inquisition	Emotional burden and additional stress due to the inquiry and investigation of the incident by the risk management department, which focused on the needs of the patient and the organization.	In most cases, no appeal was made to the nurse to demand that he/she be safe or offer support options. In cases where organizational support was offered, some nurses felt that it was illegitimate to admit weakness.	Empathy, transparency, and reliable information at the right time
Obtaining emotional first aid	Lack of SVP awareness. Lack of legitimacy of organizational support. Personal barriers.	Most of the nurses did not request proactive help.	Mostly, [14] expressed a need for professional help and the development of suitable training programs. Acknowledging the legitimacy of seeking professional support. Establishment of anonymous support sources.
Moving on	Loss of confidence and professional self-efficacy led to defensive medicine and/or over-treatment, avoidant behavior, and empathy erosion.	Mostly, no proactive contact was made to find out the nurse’s emotional state.	Establishment of well-tailored support both within and outside the organization, even as a requirement.

## Data Availability

Not applicable.

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
