# Peer review of "Nurses’ Silence: Understanding the Impacts of Second Victim Phenomenon among Israeli Nurses"

_healthcare, 2023, doi:10.3390/healthcare11131961_

Round 1

Reviewer 1 Report

Thank you for the possibility to review the manuscript “Nurse´s Silence: Understanding the Impacts of Second Victim Phenomenom Among Israeli  Nurses”.

I have carefully read this manuscript with great pleasure. The work addresses an important issue in healthcare. It is overall well-written and informative. The results are clearly described and the discussion is relevant.  Ethical approval is stated as well as informed consent. I lack information about trial registration. The references are appropriate and mostly from recent years.

Specific minor comments

Introduction

Line 41: The word “still” is repeated by mistake.

Materials and Methods

Results

Lines 291-293: Change type to Italics.

Table 2: Three reports of an error without harm. The time specifications are not correct. “Over than year” and “Over that Eight months” do not make sense. 

Discussion

Lines 419-420: The discussion concerning the “third victim” is not possible to follow, possibly due to language difficulties? Please rephrase!

Minor corrections are needed.

Author Response

Re: Manuscript ID Healthcare-2469908- Decision letter- revisions

 Dear Editor,

The authors wish to thank you and the reviewers for their important comments. We value your input and are grateful for the opportunity to improve our article. For ease reading, the responses to the reviewers' comments are arranged in a table.

We have addressed each of the comments and corrected the manuscript accordingly. The corrections were made with the "Track Changes" function, so that changes are easily visible to the editor and reviewers.

REVIEWER'S NO. 1 COMMENTS

Reviewer comments

Author response

Introduction

Line 41: The word “still” is repeated by mistake.

That was corrected.

Results

Lines 291-293: Change type to Italics.

That was corrected.

Table 2: Three reports of an error without harm. The time specifications are not correct. “Over than year” and “Over that Eight months” do not make sense. 

That was corrected.

Discussion

Lines 419-420: The discussion concerning the “third victim” is not possible to follow, possibly due to language difficulties? Please rephrase!

This sentence was changed to “ Maybe the paradox is the existence of a system that is expected to support the employee, (second victim) while,  at the same time, conduct an inquiry into the employee’s actions, and represent the best interests of itself (third victim), as well as those of the patient (first victim) (34).

Comments on the Quality of English Language

Minor corrections are needed.

The article was proofread by a professional editor and corrections were made, as needed.

Reviewer 2 Report

Dear Rinat, Yael, Inbal and Rachel,

Your work addresses an issue that is often undisclosed among  healthcare delivery professionals. I'm aware the collecting these types of data are difficult but your findings would be more compelling if you could attain an "N" of 30 so you could draw some statistical inference. I also believe your conclusion would be enhanced by adding the examples you referenced [Finney (35), Krommer et. al. (31), Rossin & Kanti (25)] within the Conclusion. Hope this helpful; I look forward to seeing your work as a publication in MDPI Healthcare.

Best,

Author Response

Re: Manuscript ID Healthcare-2469908- Decision letter- revisions

 Dear Editor,

The authors wish to thank you and the reviewers for their important comments. We value your input and are grateful for the opportunity to improve our article. For ease reading, the responses to the reviewers' comments are arranged in a table.

We have addressed each of the comments and corrected the manuscript accordingly. The corrections were made with the "Track Changes" function, so that changes are easily visible to the editor and reviewers.

REVIEWER'S NO. 2 COMMENTS

Reviewer comments

Author response

I'm aware the collecting these types of data are difficult but your findings would be more compelling if you could attain an "N" of 30 so you could draw some statistical inference.

Thank you for helpful comment. We do realize that this is a preliminary qualitative study in a field that has hardly been studied in Israel.

The purpose was to learn and gain more information about the various aspects of the phenomenon.

In a future study, we definitely intend to expand the study population in a way that will allow statistical analyses to be performed.

To clarify this point, we added a sentence to the limitations section, “In addition, this was an initial qualitative study of a topic not yet researched in Israel; we hope to pursue larger mixed methods studies in the future, as awareness of this phenom-enon grows among healthcare professionals.”

I also believe your conclusion would be enhanced by adding the examples you referenced [Finney (35), Krommer et. al. (31), Rossin & Kanti (25)] within the Conclusion.

To enhance our conclusion, we took your suggestion, and added to the conclusion with the following sentences, “Programs that have taken these aspects under consideration have proven successful in other countries. For example, in an anesthesiology department in the United States, a three-tiered support program was implemented; over 90 staff members requested some level of support in an anonymous fashion.  Of those who received support, the large majority felt that the help offered by trained peers had been very beneficial (35). A descriptive study in Austria asked participants in one large healthcare facility to rank theoretical support options.  The option chosen most often was receiving legal advice after an AE, followed by increased access to professional counseling (31). In Israel, the nursing administration took the initiative to educate supervising staff about a more supportive approach to managing medication errors. Their analysis found that overall, nursing staff felt more supported after such an event, and there was less fear of punishment. These results demonstrate the possibility of raising awareness and changing the organizational culture (25). Developing organizational resources and implementing support programs for those experiencing SVP within an organization, raising awareness of SVP among management, and building dedicated training programs for providers and peers are all necessary actions.”

Reviewer 3 Report

1. The purpose of this paper is to examine the second victim phenomenon among Israeli nurses involved in adverse events in order to achieve better policies to improve the quality of care and patient safety.

2. By examining the second victim phenomenon among Israeli nurses, policymakers and healthcare administrators can gain valuable insights into the challenges faced by healthcare providers in the aftermath of adverse events. This knowledge can inform the development of policies and interventions aimed at supporting nurses and mitigating the negative consequences they may experience. The above suggests that the research topic is relevant.

3. This study examined the experiences of 15 nurses, based on which the Authors tried to map out gaps between the organisational support that these nurses had received and the organisational support that was desired.

4. The text of the paper is well structured, as is the abstract. The purpose stated by the Authors has been achieved. However, the scientific approach used in the study is not unambiguous.

5. The paper is a promising piece of work, but it lacks more sophisticated research methods. Basing all the observations only on in-depth interviews raises doubts about the quality of the scientific analysis.

The sample proposed by the Authors is questionable. The Authors based their study on fifteen in-depth interviews, whereas when reviewing the literature, they themselves indicate that similar studies have "analysed the responses of 150 nurses", which allows for more reliable results.

6. The literature cited in the work corresponds to the current research status in the field of the discussed issues.

Author Response

Re: Manuscript ID Healthcare-2469908- Decision letter- revisions

 Dear Editor,

The authors wish to thank you and the reviewers for their important comments. We value your input and are grateful for the opportunity to improve our article. For ease reading, the responses to the reviewers' comments are arranged in a table.

We have addressed each of the comments and corrected the manuscript accordingly. The corrections were made with the "Track Changes" function, so that changes are easily visible to the editor and reviewers.

REVIEWER NO. 3 COMMENTS

Reviewer comments

Author response

The text of the paper is well structured, as is the abstract. The purpose stated by the Authors has been achieved. However, the scientific approach used in the study is not unambiguous.

As we hoped that we clarified in the methods section, we chose this method of analysis as an initial way to obtain a wide and authentic picture of the situation.  We state that “We targeted participants with relevant knowledge and experience regarding the study's aim of investigating nurses' perceptions, behaviors, attitudes, and coping strategies regarding SVP. The sample size was based on data saturation until no new information was obtained from further participants.” These methods are widely supported in the literature and we bring two sources to support our choice  ( Streuber, H.J.; Carpenter, D.R. Qualitative research in nursing: Advancing the humanistic imperative. 3rd ed. China: Library of Congress Cataloging-in Publication Data, 2020.  Munhall, P.L. (Ed.) Nursing research: A qualitative perspective (3rd ed.) National League for Nursing. Jones & Bartlett, Boston, 2001.

We also added a sentence to further explain our future goals in the limitations section, “In addition, this was an initial qualitative study of a topic not yet researched in Israel; we hope to pursue larger mixed methods studies in the future, as awareness of this phenom-enon grows among healthcare professionals.”

The paper is a promising piece of work, but it lacks more sophisticated research methods. Basing all the observations only on in-depth interviews raises doubts about the quality of the scientific analysis.

The sample proposed by the Authors is questionable. The Authors based their study on fifteen in-depth interviews, whereas when reviewing the literature, they themselves indicate that similar studies have "analysed the responses of 150 nurses", which allows for more reliable results.

The purpose of this study was to learn and gain information about the various aspects of the SVP among nurses in Israel.

In a future study, we definitely intend to expand the study population in a way that will allow statistical analyses to be performed, using the mixed methods approach.

Additionally, we have clarified the results of the only mixed methods study conducted in Israel thus far on this topic, demonstrating that that study, too, had a small sample for the qualitative section of their study. “One mixed methods study analyzed responses of nurses working in internal medicine departments, to understand the effects of patient's suicidal attempts and explore the association between these experiences and nurse absenteeism and turnover. Seven nurses contributed to the qualitative analysis, and identified feelings of isolation, helplessness, and loneliness after the incident.  Within the quantitative analysis, the researchers identified a correlation between second victim distress, loneliness absenteeism and turnover. Higher levels of emotional distress were also negatively correlated with organizational support.” (18)